# Visual field map clusters in human frontoparietal cortex

**Wayne E Mackey[1], Jonathan Winawer[1,2], Clayton E Curtis[1,2]\***

[1]Center for Neural Science, New York University, New York, United States;
[2]Department of Psychology, New York University, New York, United States

**Abstract** The visual neurosciences have made enormous progress in recent decades, in part because of the ability to drive visual areas by their sensory inputs, allowing researchers to define visual areas reliably across individuals and across species. Similar strategies for parcellating higher-order cortex have proven elusive. Here, using a novel experimental task and nonlinear population receptive field modeling, we map and characterize the topographic organization of several regions in human frontoparietal cortex. We discover representations of both polar angle and eccentricity that are organized into clusters, similar to visual cortex, where multiple gradients of polar angle of the contralateral visual field share a confluent fovea. This is striking because neural activity in frontoparietal cortex is believed to reflect higher-order cognitive functions rather than external sensory processing. Perhaps the spatial topography in frontoparietal cortex parallels the retinotopic organization of sensory cortex to enable an efficient interface between perception and higher-order cognitive processes. Critically, these visual maps constitute well-defined anatomical units that future studies of frontoparietal cortex can reliably target.

## Introduction

A fundamental organizing principle of sensory cortex is the topographic mapping of stimulus dimensions (*Mountcastle, 1957*; *Kaas, 1997*). For instance, visual areas contain maps of the visual field, wherein the spatial arrangement of an image is preserved such that nearby neurons represent adjacent points in the visual field (*Inouye, 1909*; *Holmes, 1918*). At a larger scale, multiple visual field maps are arranged in clusters, in which several adjacent polar angle representations share a common eccentricity representation (*Wandell et al., 2005*, *2007*; *Kolster et al., 2009*). These clusters are thought to form larger, more efficient processing units by sharing computational resources and minimizing the length of axons connecting the portions of the maps with similar spatial receptive fields (*Wandell et al., 2005*). Furthermore, it has been suggested that clusters, rather than individual visual field maps, organize specializations in cortical function (*Bartels and Zeki, 2000*). To date, more than 20 visual field maps have been identified in the human brain, several of which are organized into clusters (*Wandell et al., 2005, 2007*; *Larsson and Heeger, 2006*; *Wandell and Winawer, 2011*; *Arcaro and Kastner, 2015*; *Wang et al., 2015*; *Barton and Brewer, 2017*).

Recently, using modified versions of the standard traveling wave method for mapping early visual cortex, several labs have identified representations of polar angle along the intraparietal sulcus (IPS) in posterior parietal cortex and the precentral sulcus (PCS) in frontal cortex (*Sereno et al., 2001*; *Schluppeck et al., 2005*; *Silver et al., 2005*; *Kastner et al., 2007*; *Swisher et al., 2007*; *Jerde et al., 2012*). Not surprisingly, these maps have attracted a great deal of attention given their presumed involvement in a wide range of cognitive and sensorimotor processes, including attention, working memory, and decision-making (*Posner et al., 1984*; *Wilkins et al., 1987*; *Bechara et al., 1994*; *Manes et al., 2002*; *Mackey et al., 2016a*, *2016b*). Some of these visual maps may correspond to the human homologs of well-characterized areas in the macaque brain that are

**\*For correspondence:** clayton. curtis@nyu.edu

**Competing interests:** The authors declare that no competing interests exist.

topographically organized, like the lateral intraparietal area (LIP) and the frontal eye field (FEF). Yet, such inter-species homology and how these maps contribute to different aspects of behavior and cognition remain unknown. This is likely due to our limited understanding of the basic organizing principles of these maps. Further challenges are posed because the maps in parietal and especially frontal cortex have less reliable stimulus-evoked BOLD signals compared to those in visual cortex, are more coarsely organized, and show less consistent topography across subjects. In some cases, it is uncertain as to whether a region merely has a contralateral bias as opposed to containing an actual topographic map (*Hagler and Sereno, 2006*; *Kastner et al., 2007*; *Silver and Kastner, 2009*; *Jerde et al., 2012*; *Patel et al., 2014*). Here, we take a step back from trying to understand the functions and instead systematically characterize the basic organizing principles of these putative visual field maps in frontoparietal cortex.

Establishing the organization of maps in parietal and frontal cortex will have several major impacts. First, the identification and better characterization of which maps are human homologs of macaque areas will facilitate better translation of non-human primate models of human cognition to humans. Second, understanding the topography of parietal and frontal maps will enable researchers to aggregate results at the level of individual maps or even small areas within maps, rather than at the level of large regions of cortex, similar to the successes in delineating maps in occipital cortex in animals (*Essen and Zeki, 1978*; *Desimone and Ungerleider, 1986*; *Gattass et al., 2005*) and humans (*Sereno et al., 2001*; *Engel et al., 1997*). Indeed, the fundamental reason the *visual* neurosciences have outpaced the *cognitive* neurosciences is the ability to reliably define and to study the function of the same areas across individuals and across labs. Such convention facilitates comparisons between studies of different computations and representations across different subject populations and methods of measurement. This effectively creates a worldwide, across-time, collaboration between all labs. As such, the organization of visual field maps in early visual cortex has been well characterized, to the point of enabling the development of detailed templates of the visual field for V1–V3 (*Dougherty et al., 2003*; *Benson et al., 2012*, *2014*). By contrast, much less is known about the organization of visual field maps in frontoparietal cortex, and thus presents a critical roadblock for understanding their functions.

To these ends, we focus on characterizing the organization and retinotopic properties of putative visual field maps in frontoparietal cortex. We estimated population receptive field (pRF) parameters in topographic areas in early visual, parietal, and frontal cortices. The pRF method not only estimates a voxel's polar angle and eccentricity preference, but also its receptive field (RF) size, and has been shown to map topography more accurately than conventional traveling wave methods (*Dumoulin and Wandell, 2008*). However, two important challenges exist when attempting to identify visual field maps in frontoparietal cortex. First, passive viewing of high-contrast spatial patterns elicits weak and non-systematic responses in frontoparietal cortex (*Silver et al., 2005*; *Saygin and Sereno, 2008*). Therefore, identifying visual field maps in higher order frontoparietal cortex requires more cognitively demanding stimulation that taxes attention or memory (*Silver et al., 2005*; *Jerde et al., 2012*). Second, the large RF sizes expected in frontoparietal cortex make it difficult for linear RF models to accurately characterize response properties to stimuli that vary in size. To overcome these challenges, we developed a novel, attention-demanding task specifically designed to elicit robust, systematic responses in frontoparietal cortex (*Figure 1A*). Moreover, we estimated pRFs with a model that accounts for nonlinear responses to stimuli of varying size (*Kay et al., 2013*; *Winawer et al., 2013*). Nonlinear spatial summation is more pronounced in extrastriate maps than in V1 and is likely to be even more so in frontoparietal cortices.

## Results

During a single functional neuroimaging session, observers swept their focus of attention across the visual field, while maintaining central fixation, in order to perform a difficult motion discrimination task. The task was designed to tax attentional resources that are presumably controlled by activity in topographic maps in frontoparietal cortex (*Figure 1A*, *Video 1*). A bar aperture swept across the visual field in discrete steps, vertically or horizontally traveling in four possible directions: left to right, right to left, top to bottom, bottom to top. The bar was comprised of three rectangular patches, each of which contained a random dot kinematogram (RDK) moving in a particular direction. The central patch contained 100% coherent motion and the two flanking patches contained

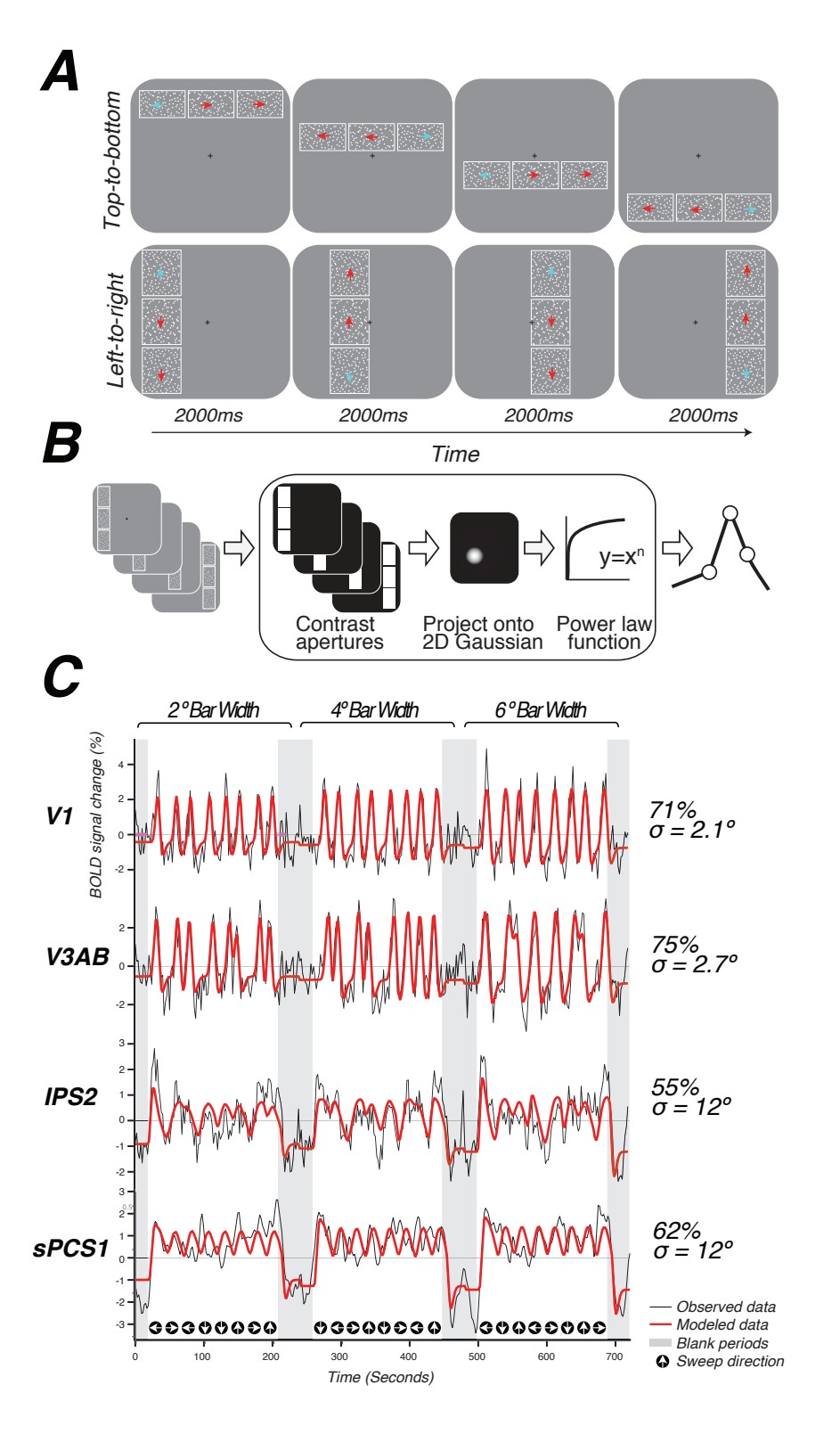

**Figure 1.** Topographic mapping and population receptive-field modeling. (**A**) The discrimination task used for topographic mapping. Subjects fixated at the center of the screen while attending covertly to a bar composed of three apertures of moving dots incrementally traversing the screen. Subjects indicated on each trial which aperture (left, right, top, or bottom) was comprised of dots whose motion direction matched that of the dots in the middle sample aperture. Motion coherence was staircased in order to tax attention constantly. The white outlines around each of the three apertures are

*Figure 1 continued on next page*

*Figure 1 continued*

shown here for clarity, but were not visible to subjects. (**B**) Schematic of the nonlinear population receptive-field modeling procedure. Trial sequences were converted into 2D binary contrast apertures and projected onto a 2D Gaussian representing a predicted pRF. A static non-linearity was applied to account for compressive spatial summation. (**C**) Example model fits from single voxels in multiple visual field maps. pRF model predictions are shown in red, actual data for an individual voxel for a given visual field map are shown in black. Stimulus sweep direction and bar width are shown above and below the model fits. Estimated pRF size and variance explained for each voxel are shown to the right.

low coherence motion. Observers pressed a button to indicate which of the two flanking patches (above or below for vertical bars, left or right for horizontal bars) matched the RDK direction of the middle patch. The motion coherence of the flanking patches was staircased to ensure the task remained difficult throughout the duration of the scanning session (75% accuracy).

We used a nonlinear pRF model to predict the BOLD response of each individual voxel to the visual stimulus (*Figure 1B*). Each model is specified by the center (*x,y*) and size (σ, or standard deviation of an isotropic 2D Gaussian) of the pRF, and uses a power-law exponent (*n*) to account for subadditive spatial summation (*Equations 1 and 2*). The non-linearity interacts with the Gaussian standard deviation to make an effective pRF size of $\sigma/\sqrt{n}$ (*Kay et al., 2013*; *Winawer et al., 2013*). We excluded voxels from further analysis if less than 10% of the variance in the time series was explained by the pRF model (*Equation 3*). We also excluded voxels with pRF centers outside of the limits of our visual display (12 degrees of visual angle). The pRF fitting strategy proposed by *Dumoulin and Wandell (2008)* was a coarse-to-fine approach, in which the initial coarse fit was solved on time series that were spatially blurred (approximating a Gaussian kernel of 5 mm width at half height) and temporally decimated (2x), and that used gridded parameters rather than searching for the best fit using nonlinear search optimization algorithms; the second stage then used the solution of the grid fit as a seed for a search, and applied this search to the unblurred (in space and time) time series. Here, we used only the first stage (the grid fit), applied to the smoothed and temporally decimated time series. The grid fit is more robust to noise (although also less accurate when noise is low). Our primary goal was to map frontoparietal cortex, where pRFs were expected to be larger, and hence the stimulus relevant signals were expected to be dominated by lower temporal frequencies compared to those of early visual cortex, where pRFs are much smaller. The temporal decimation in the grid fit reduces sensitivity to high temporal frequency noise. Furthermore, in frontoparietal cortex, stimulus-related signals were expected to be less reliable than those in the visual cortex; as a result, the spatial blurring in the grid fit was beneficial. Subsequently, we performed the second stage (search fit) to ensure that our results were not an artifact of artificial structure imposed by the grid fit due to interpolating voxels. We found no discernible differences between the grid and search fits (See *Model reliability and comparisons* below). The grid included the same set of possible values in the solution as those described by *Dumoulin and Wandell (2008)*, with the addition of the power law exponent, which was gridded to be 0.25, 0.5, 0.75, or 1. A value of 1 indicates a linear fit; values smaller than 1 indicate increasingly more sub-additive spatial summation.

We identified retinotopic maps in early dorsal visual areas (V1, V2, V3, V3A&B) as well as at least four maps along the intraparietal sulcus (IPS0, IPS1, IPS2, IPS3) and two regions with spatial tuning along the precentral sulcus in frontal cortex in each hemisphere of all subjects. The superior portion of the precentral sulcus region contained two distinct visual field maps that we refer to as sPCS1 and sPCS2. Example model fits

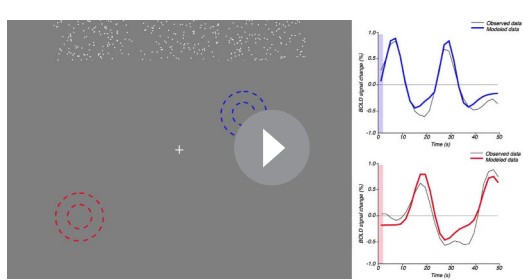

**Video 1.** Example population receptive fields estimated for two voxels. As the attended stimulus sweeps across the visual field, it evokes a spatially and temporary selective response. The red and blue dashed circles represent the position of two pRFs from the right and left V1, respectively. The radius of the circle depicts 1 and 2 standard deviations of the Gaussian-shaped pRF. The BOLD time courses and the model fits are shown to the right with a moving bar that synchronizes the stimulus video and time courses.

for voxels in different visual field maps are shown in *Figure 1C*. Although we were able to observe additional parietal maps in some subjects, we restricted further analysis to maps consistently observed in both hemispheres of every subject. All areas are described in further detail below.

## Visual cortex

As an important control, our methods revealed the functional organization and pRF properties typically observed in early visual cortex. Polar angle and eccentricity representations revealed the expected patterns in V1, V2d, V3d, and V3A&B (*Figure 2A and B*; *Figure 2—figure supplement 1*). The progression of these areas begins medially from the occipital pole in the calcarine sulcus with V1, and extends dorsally and laterally along the surface of the cortex. V1 contains a continuous angular representation of the contralateral hemifield, beginning at the upper vertical meridian (UVM) on the lower bank of the calcarine sulcus, and progressing to the lower vertical meridian (LVM) along the inferior-to-superior direction. V2 sits adjacent to the LVM border of V1, and begins a reversal of polar angles representing half of the contralateral hemifield and progressing to the horizontal meridian (HM). V3 sits adjacent to the HM border of V2, beginning another angle reversal back towards the LVM, and also represents half of the contralateral hemifield. Finally, V3A begins along the border of V3 in the periphery (but not fovea), and contains a full and continuous angular representation of the contralateral hemifield from the LVM to the UVM. V3B is adjacent to V3A, divided by a shared foveal representation, as reported previously (*Press et al., 2001*; *Wandell et al., 2005*).

Together, these maps form two distinct visual field map clusters. A cluster is comprised of a group of angle representations that all share a confluent fovea (*Wandell et al., 2005*; *Kolster et al., 2009*). Within a cluster, the boundaries of adjacent angle representations can be defined by reversals in polar angle progression or eccentricity. For example, the boundary between V1 and V2 is identified by a polar angle reversal at the LVM, as described above. V1, V2, and V3 share a common foveal representation centered near the occipital pole that extends towards the collateral sulcus. V3A and V3B comprise a second cluster because they share another foveal representation, anterior to and distinct from the foveal representation shared by V1, V2, and V3. They are divided by an eccentricity reversal rather than by a polar angle reversal.

Moving up the hierarchy of early visual cortex, from V1 to V3A&B, pRF parameters begin to differ systematically. For example, pRF sizes at a given eccentricity generally increased across visual field maps, and within each visual field map in visual cortex, pRF size increased monotonically with eccentricity (*Figure 2C*). Consistent with previous studies investigating spatial summation in pRFs (*Kay et al., 2013*), we found that in extrastriate maps, the pRF models show more sub-additive spatial summation, indicated by a smaller power law exponent. The fraction of voxels with the minimal pRF exponent allowed by our grid search (0.25) was least in V1 (33%), and substantially higher in extrastriate areas (68% in V2; 80% in V3; 77% in V3A&B).

## Parietal cortex

We found clear patterns of systematic organization in eccentricity and polar angle representation in the intraparietal sulcus (*Figure 3*; *Figure 3—figure supplement 1*). Beginning with IPS0 and progressing through IPS3, each map contains a full representation of the contralateral hemifield, while the polar angle reversals demarcate the boundaries of each individual map. Additionally, each successive map lies anterior to the visual field map before it. Starting with the most posterior map, IPS0 lies at the intersection of the parietal-occipital sulcus and the intraparietal sulcus, adjacent to the UVM representation of V3A&B. The angular representation systematically progresses from the UVM to the LVM, where another angle reversal takes place, creating the posterior border of IPS1. The angular gradient of IPS1 sweeps from the LVM back towards the UVM, where it borders IPS2. IPS2 then contains an angular progression from the UVM to the LVM, where it borders IPS3. IPS3 then contains an angular progression from the LVM to the UVM.

Using these novel methods, we discovered that maps along the IPS are organized into two clusters of visual maps. IPS0 and IPS1 share a confluent fovea, while IPS2 and IPS3 share another distinct foveal representation (*Figure 3*). Notably, we find that the direction of the eccentricity gradient shared by IPS0 and IPS1, as well as that shared by IPS2 and IPS3 is the same. In both clusters, there is a mediolateral representation of eccentricity, where the foveal region is found near the fundus of the intraparietal sulcus, and then parafoveal-to-peripheral representations progress medially towards

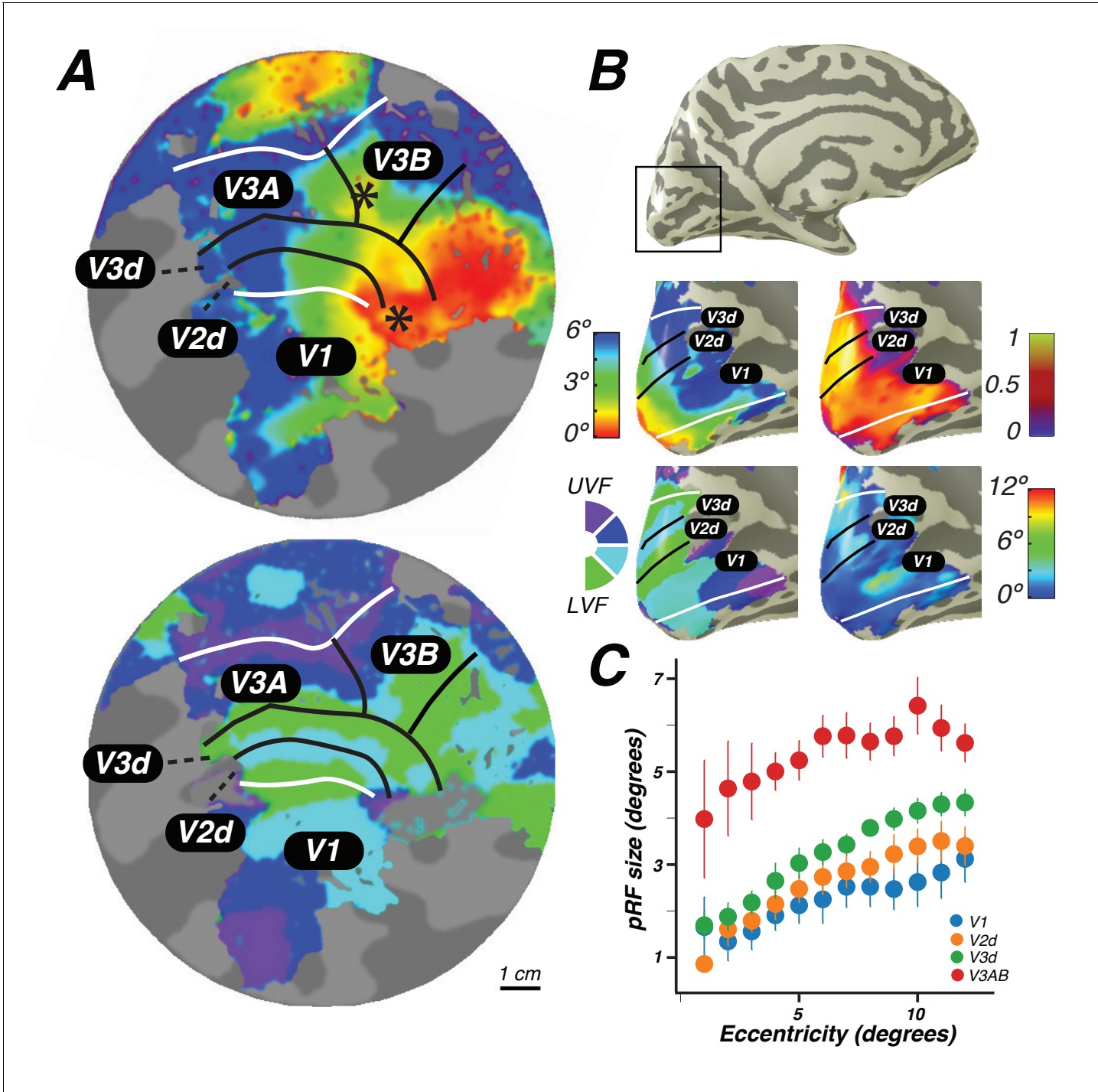

**Figure 2.** Visual field maps in early visual cortex. The color of each voxel indicates the best fit pRF parameter for the data being displayed. (A) Eccentricity (top) and polar angle (bottom) maps in the left hemisphere of an example subject projected onto a flattened representation of the cortical surface, where dark gray denotes sulci and light gray denotes gyri. V1, V2, and V3 share a confluent fovea while V3A and V3B share another. (B) Eccentricity (top left) and polar angle (bottom left) maps, along with variance explained (top right) and pRF size (bottom right) of an example subject projected onto an inflated cortical surface. (C) Relationship between pRF size and eccentricity. pRF sizes of voxels in V1, V2, V3, and V3AB increase with eccentricity. Error bars represent ±1 SEM across subjects.

The following figure supplement is available for figure 2:

**Figure supplement 1.** Individual subject visual cortex data.

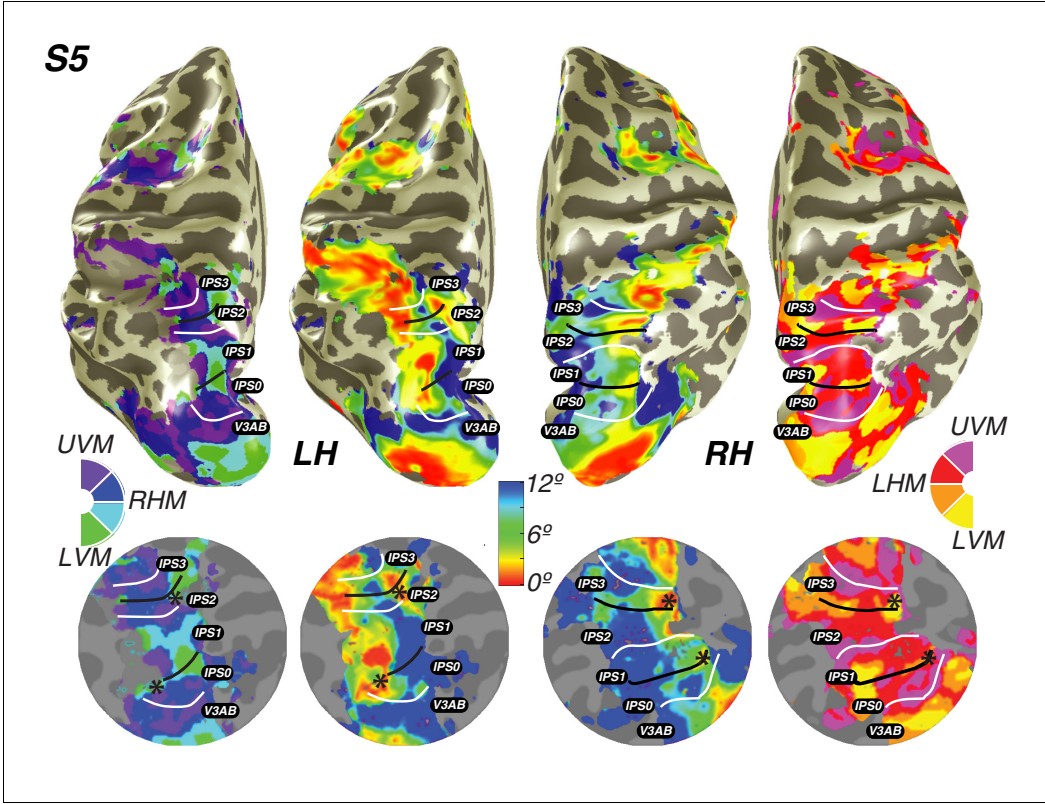

**Figure 3.** Visual field maps in parietal cortex. Maps of polar angle and eccentricity on an inflated cortical surface (top) and on a flattened representation of the cortical surface (bottom) for an example subject. IPS0/IPS1 form one visual field map cluster, while IPS2/IPS3 form another. Each cluster consists of two angle maps that share a confluent foveal representation. White lines denote the boundaries at the upper vertical meridian (UVM) and black lines denote the lower vertical meridian (LVM); asterisks denote foveal representations.

The following figure supplement is available for figure 3:

**Figure supplement 1.** Individual subject parietal cortex data.

---

the medial wall (*Figure 3—figure supplement 1*). However, unlike maps in early visual cortex, parietal visual field maps did not allow us to measure pRF sizes with sufficient accuracy. Estimates of pRF size for nearly all voxels in parietal visual field maps were ~12 degrees, even for voxels with pRF centers at the fovea. As 12 degrees is both the maximum stimulus extent in our experiment and the upper boundary of our grid fit, it is likely that these pRF size estimates are a floor on the true size, rather than an accurate measure of the pRF size of voxels in this region. However, these measurements do indicate that the sizes of pRFs in parietal cortex are large, and probably larger than 12 degrees.

Maps in parietal cortex, like extrastriate maps in visual areas, showed a systematic sub-additivity in spatial summation. The vast majority of voxels were best fit by a pRF model with the minimal pRF exponent allowed by our grid (IPS0: 73%; IPS1: 75%; IPS2: 81%; IPS3: 89%). These results are comparable to the low exponent found in measurements of ventral occipitotemporal face-selective regions (0.20, 0.16, 0.23 in three face-selective ROIs) (*Kay et al., 2015*).

## Frontal cortex

We also discovered that visual field maps in frontal cortex are organized by polar angle as well as by eccentricity. Although polar angle representations have been described before (*Kastner et al., 2007*; *Saygin and Sereno, 2008*; *Jerde et al., 2012*), representations of eccentricity have remained undiscovered until now. Two regions of frontal cortex along the PCS exhibited spatial tuning: one in

the superior portion (sPCS) and another in the inferior portion (iPCS). Each region contains a gradient of eccentricity with a central foveal representation that radiates towards the periphery. In the sPCS, the fovea is represented in the fundus of PCS where it intersects the superior frontal sulcus. This organization was consistent across both hemispheres in all subjects (*Figure 4*). The discovery of the foveal representation in a location common across subjects allowed us to separate the region into two visual field maps, identifiable in each subject. We refer to these new maps as sPCS1 and sPCS2, consistent with naming visual field maps according to their anatomical location (*Larsson and Heeger, 2006*; *Arcaro et al., 2009*). Both sPCS1 and sPCS2 contain a full and continuous representation of the contralateral hemifield (*Figure 5*; *Figure 5—figure supplement 1*). The superior border of sPCS1 begins at the LVM and continues to the UVM, where an angle reversal occurs, signaling the border of sPCS2. sPCS2 then contains an angular progression from the UVM back to the LVM. The inferior portion of the PCS (iPCS) also contained spatial representations and was distinct from the sPCS cluster (*Figure 5*; *Figure 5—figure supplement 2*). The foveal representation in the iPCS is located at the junction of the PCS and the inferior frontal sulcus. Although the topography in the iPCS region was not sufficiently regular across subjects to propose a generalized map schema, the eccentricity representation was regular in that peripheral representations typically surrounded a foveal representation. Further, while the map organization was less clear than the sPCS maps, both in terms of consistency across subjects and structure within a subject, the visual field coverage nonetheless showed a systematic representation of the contralateral visual field, similar to sPCS1 and sPCS2, which we return to in the next section (*Visual field coverage density and laterality*). Moreover, the iPCS maps were as accurate as the sPCS maps in terms of cross-validated prediction error and were also as lateralized, as we describe below (*Model reliability and comparison*, *Visual field coverage laterality and density*). These results indicate that the iPCS map, like the sPCS maps, contains a clear representation of spatial information.

As was the case in parietal cortex, in frontal cortex, estimates of pRF size were large and pRF exponents small for nearly all voxels. Size estimates were limited by the field of view of our display

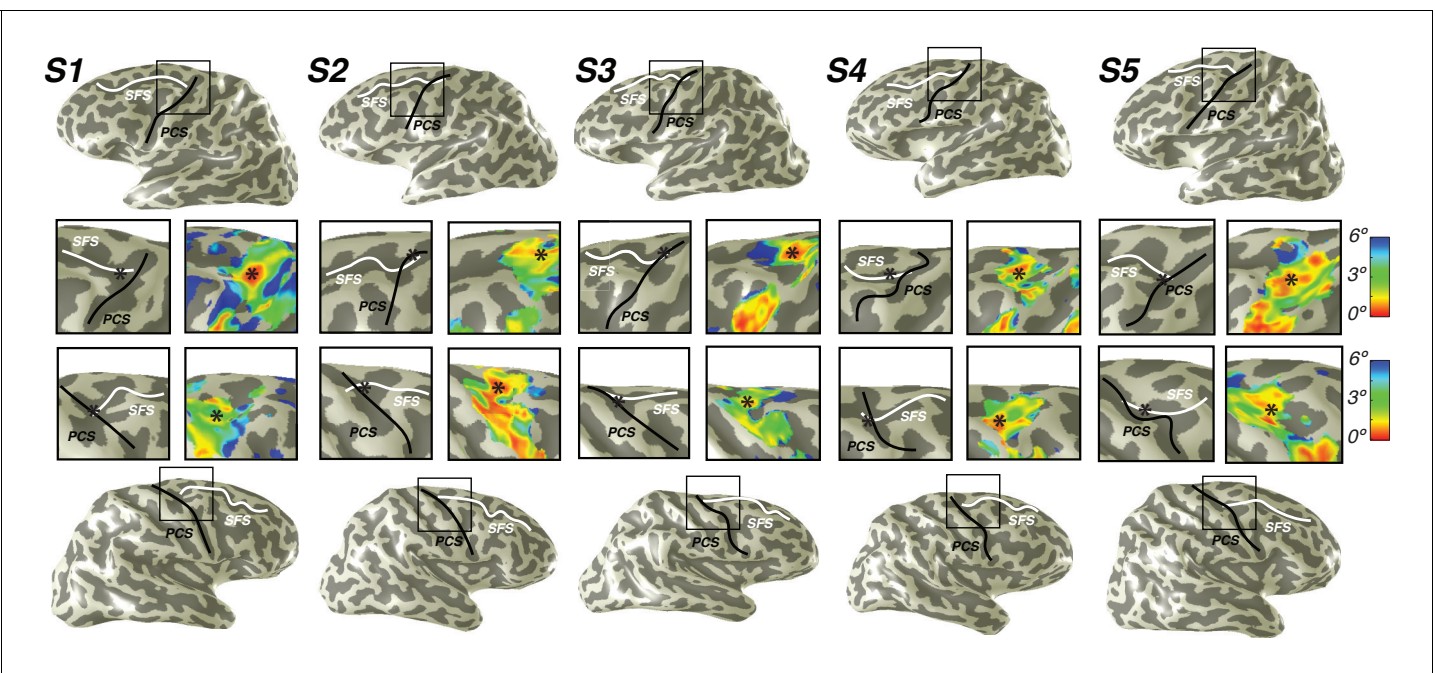

**Figure 4.** Representation of fovea in sPCS map. Left (top) and right (bottom) hemispheres for each individual subject are shown. Top and bottom rows mark the anatomical locations of the superior frontal sulcus (SFS: white line) and the superior precentral sulcus (sPCS: black line) on an inflated cortical surface representation of each hemisphere. For clarity, black squares represent a zoomed in view of the anatomical intersection of the SFS and sPCS. The location of the fovea (asterisk) is shown both on the anatomy (left) and on the eccentricity map (right) for each individual subject. Notice how the fovea lies at the intersection of the SFS and sPCS for each subject.

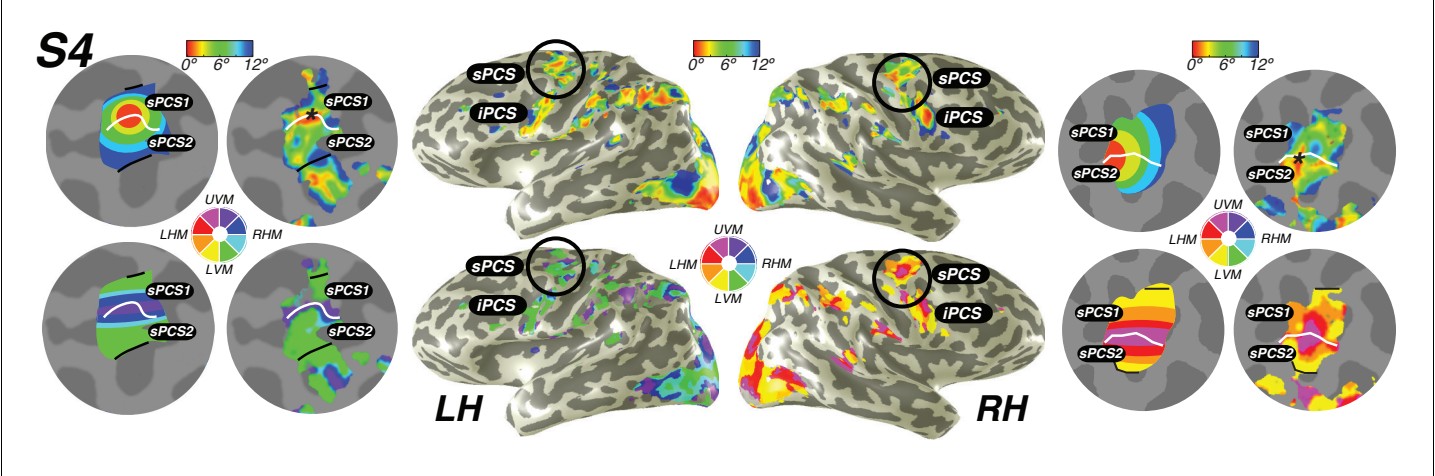

**Figure 5.** Visual field maps in frontal cortex. Maps of eccentricity and polar angle are displayed for an example subject projected on both inflated (inside) and flattened (outside) cortical surfaces. In order to demonstrate the systematic organization of each map clearly, each pair of flattened cortical surfaces depicts a cartoon schematic of the organization of each map (left flat patch) next to actual map data (right flat patch). sPCS1 and sPCS2 form a visual field map cluster, sharing a foveal representation that sits at the intersection of the PCS and the SFS. White lines denote the boundaries at the upper vertical meridian (UVM) and black lines denote the lower vertical meridian (LVM). The foveal representation in iPCS sits at the intersection of the PCS and the IFS.

The following figure supplements are available for figure 5:

**Figure supplement 1.** Individual subject superiorprecentral sulcus(sPCS) maps.

**Figure supplement 2.** Individual subject inferior precentral sulcus (iPCS) maps.

and the analysis method. Most pRFs in frontal cortex had a size estimate at the upper bound of our analysis (12 deg) and an exponent at the lower bound (0.25; iPCS: 78%; sPCS1: 82%; sPCS2: 75%). As discussed in the next two sections, the large pRF size does not indicate a failure to estimate spatial tuning, as indicated by the cross-validated model fits and the laterality indices. Nonetheless, future studies with larger displays are probably needed to estimate pRF size in these regions accurately.

## Visual field coverage laterality and density

A pRF model summarizes the sensitivity of a single cortical site (for example, a voxel) to positions in the visual field. By combining the pRFs across sites within a region of interest, one can visualize the field of view of the region of interest, also called the visual field coverage (*Amano et al., 2009*; *Winawer et al., 2010*; *Wandell and Winawer, 2015*). The visual field coverage is typically computed as the envelope of the pRFs within an ROI. Here, we took the mean of the pRFs rather than the envelope, thereby scaling the visual field coverage by the density of pRFs at any particular location in the visual field (cortical magnification), which we refer to as the coverage density. In the V1-V3 maps, the coverage density plots show a nearly total contralateral bias, as well as bias toward the fovea over the periphery (*Figure 6A*). As we only estimated pRF models in the dorsal visual maps of V2 and V3, coverage was limited to the lower quadrant of the contralateral visual field as expected. The contralateral and foveal biases in early visual cortex reflect three aspects of the pRF models: (1) the centers are in the contralateral visual field, (2) the pRF sizes are relatively small, and (3) the largest number of pRF centers are close to the fovea. In V3A&B, the coverage density is wider and extends slightly into the ipsilateral visual field, as the result of larger pRF sizes. This observation was used previously to distinguish area MST from area MT in the human visual system (*Huk et al., 2002*; *Amano et al., 2009*). In parietal cortex (IPS0-IPS3) and frontal cortex (iPCS and sPCS1&2), the coverage is also centered in the contralateral hemifield, but increasingly extends into the ipsilateral field, again reflecting the larger pRF sizes. Similar to visual cortex, the coverage density is highest near the

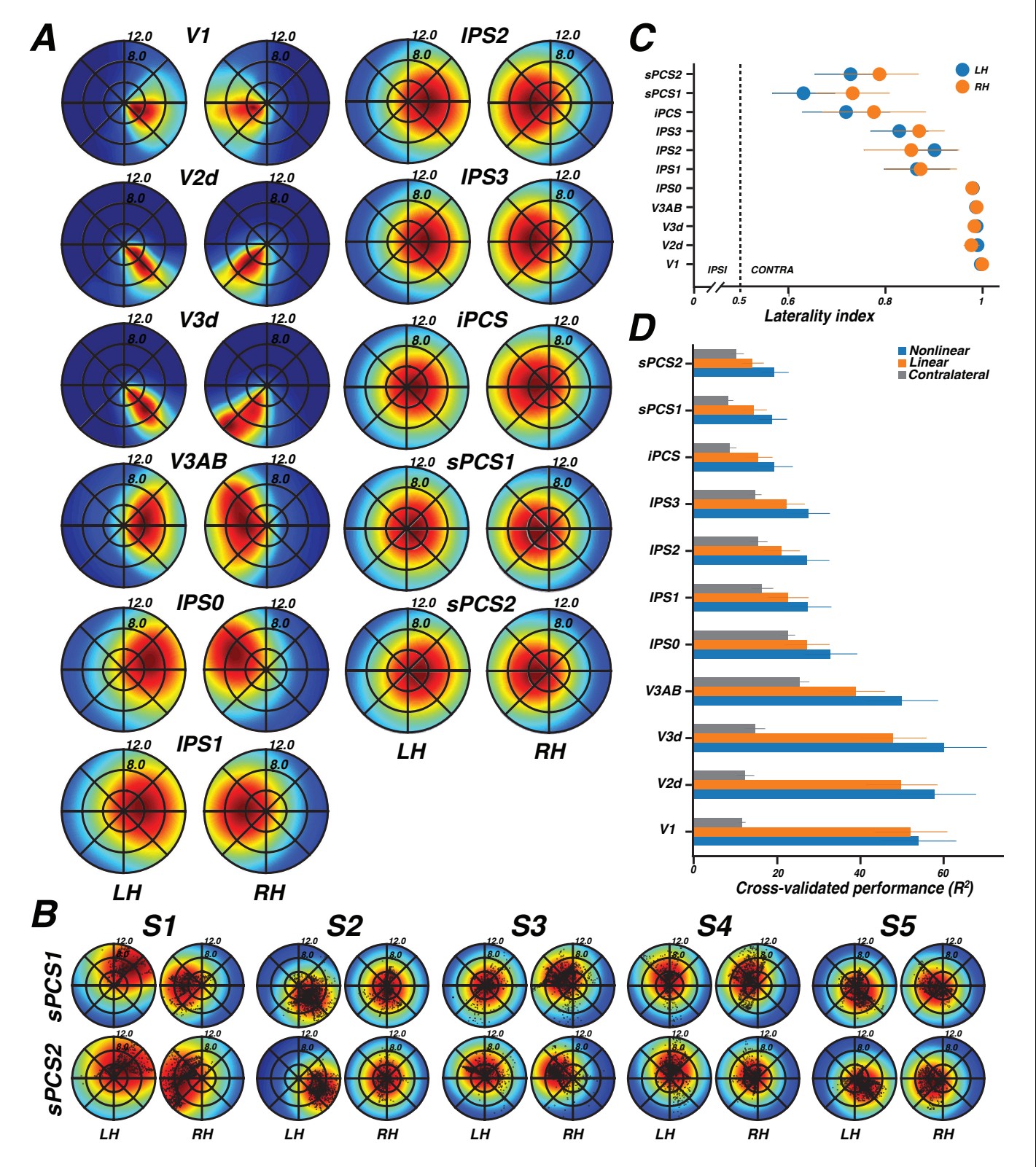

**Figure 6.** Visual field map coverage, laterality, and model comparison. (**A**) Visual field map coverage plots. While all visual field maps primarily represent the contralateral hemifield, maps in association cortices also begin to represent small portions of the ipsilateral hemifield perifoveally. This is due to the fact that pRFs are less eccentric and larger in frontal and parietal cortex than in early visual cortex. (**B**) Visual field coverage plots for the PCS in individual subjects. Each small black dot represents the center of a voxel's pRF. (**C**) Laterality index (means ± SEM across subjects). The index ranges

*Figure 6 continued on next page*

*Figure 6 continued*

from 0 (completely ipsilateral) to 0.5 (no laterality) to 1 (completely contralateral). All areas are highly contralateralized. (D) Comparison of cross validation results by model (means ± SEM across subjects). For every visual field map, the non-linear model explained the largest amount of variance, followed by the linear model, and finally the contralateral model.

fovea, indicating a qualitative similarity in cortical magnification between frontoparietal cortex and visual cortex. The visual field coverage is evident at the individual level as well as the group level, and is supported by viewing the pRF centers as well as the coverage density (*Figure 6B*).

We quantified the degree to which visual areas have lateralized pRFs using a laterality index (*Equation 4*). In agreement with the coverage density plots, the laterality index shows that early visual cortex is highly lateralized, whereas successive maps from parietal to frontal cortex become less and less lateralized (*Figure 6C*). Additionally, there is larger subject-to-subject variability in the lateralization index in parietal and frontal areas compared to visual cortex. In summary, although maps in frontal and parietal cortex have very large pRFs compared to those in visual cortex, the spatial tuning is sufficiently reliable to show clear lateralization, complete hemifield visual coverage, and maps that are organized in an orderly topographic manner.

## Model reliability and comparison

Reliably mapping topography in frontal and parietal cortex depends on nonlinear models of pRFs. We compared performance of the nonlinear pRF model to two other models: a linear pRF model (*Dumoulin and Wandell, 2008*) and a simple contralateralized response model. Comparisons with the linear model allowed us to see how much, if any, improvement in accuracy was gained by allowing for sub-additive spatial summation. Comparing with the lateralized response model allowed us to investigate whether our results indicate systematically organized maps as opposed to noisy representations of lateralized responses, with no spatial tuning other than a preference for the contralateral hemifield. The lateralized model predicted a uniform response amplitude whenever any portion of the stimulus was in the contralateral visual field, and zero response otherwise.

In order to compare model performance, we used a leave-one-out cross-validation procedure, which provides no advantage to models with additional parameters. The models were solved using two of the three stimulus types as training data (narrow, intermediate, or wide bars), and the remaining stimulus type as test data, iterated by leaving out each of the three stimulus types. We defined accuracy as the variance explained for the left-out data, averaged across the three cross-validation iterations. The ordinal ranking of the three models was the same in all 11 areas tested: the nonlinear model explained the greatest amount of variance, followed by the linear model, and then the lateralized response model (*Figure 6D*). The quantitative advantage of the nonlinear model over the linear model was smallest in V1, and larger in other maps. The improvement in performance from the nonlinear over the linear model demonstrates the benefits of including a parameter to estimate sub-additive spatial summation. This is particularly true for maps in frontal and parietal cortex, where pRFs are large and response nonlinearity estimations were at our measurement boundary. The contralateral model was much worse than either of the Gaussian models (linear or nonlinear) in visual cortex, a result of the relatively small pRFs in those areas, yet even in parietal and frontal areas, where pRFs were much larger, the contralateral model was worse than the non-linear pRF model in every area tested. This result strengthens the claim that the observed responses are spatially tuned, and not merely due to a non-specific preference for contralateral stimuli.

In order to investigate the reliability of newly discovered visual field map clusters in precentral sulcus and intraparietal sulcus, we performed a test-retest analysis on two subjects. Importantly, we found that visual field map structure (that is, polar angle and eccentricity gradients) were consistent across sessions in both subjects (*Figure 7*). Both the location of the fovea along the eccentricity gradient and the map borders between sPCS1 and sPCS2 were remarkably consistent across sessions. Similarly, along the intraparietal sulcus, the foveal location and angle boundaries were the same across the two sessions in both subjects. Therefore, the structure of visual field map clusters in frontoparietal cortex is stable across scanning sessions.

Although pRF accuracy differs considerably between nonlinear and linear models, it is not dependent on using the grid or search fit. In the traditional pRF fitting procedure outlined in

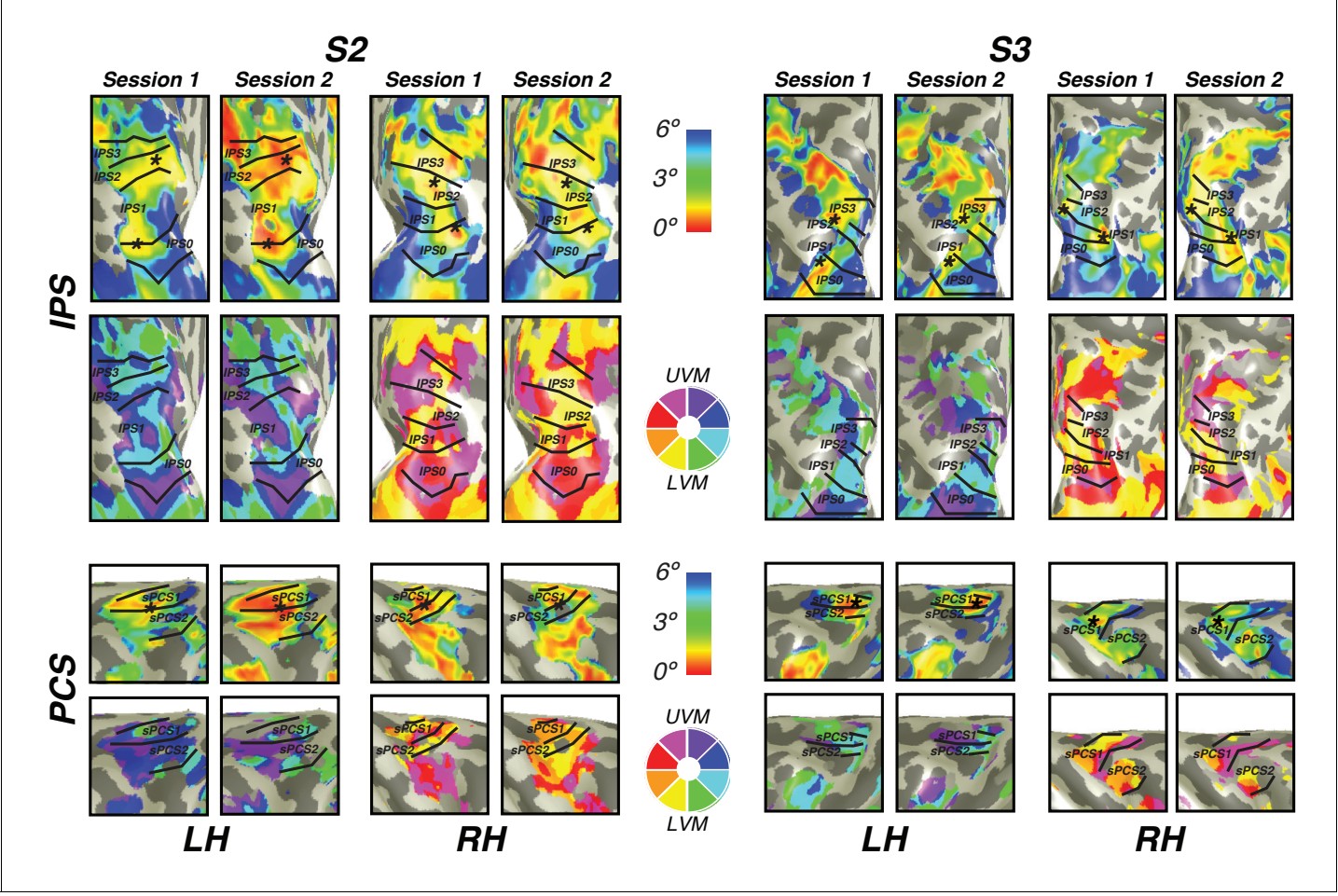

**Figure 7.** Cross-session stability of frontal and parietal visual maps. For two subjects, we show the visual maps in the intraparietal and precentral sulci derived from two independent scanning sessions. Both the eccentricity and angle representations are shown for both subjects and both hemispheres. Note the striking similarity of the maps, demonstrating that the visual map structure is stable over time and our measurement methods are reliable.

The following figure supplement is available for figure 7:

**Figure supplement 1.** Grid fit and search fit comparison.

*Dumoulin and Wandell (2008)*, pRF parameters are determined by a two-stage coarse-to-fine fitting procedure in which a grid fit is performed to generate a seed for the search fit. To investigate whether our use of the grid fit instead of the full two-stage process impacted our results, we compared our results to results from the two-stage fitting procedure. Using the same cross-validation procedure outlined above to compare nonlinear and linear models, we compared nonlinear model solutions between the grid fit and search fit. The two models—grid fit and search fit—solved on the training data both made predictions for the identical, left-out, unsmoothed test data. The grid fit performed numerically, but not statistically, better than the search fit in every ROI (*Figure 7—figure supplement 1*). Additionally, there is little to no difference in visual field map structure, polar angle gradients, or eccentricity gradients between the grid fit and search fit.

Together, these observations, coupled with our newly discovered systematic organization of frontal and parietal maps, demonstrate the existence of topographically organized visual field maps in frontoparietal cortex. Moreover, the facts that the pRF model solutions cross-validate well and are stable over time strongly indicate that the visual field maps are reliable and that our methods for measuring and modeling them are accurate.

## Positioning of frontoparietal visual topographic maps in the context of other mapping schemes

Next, we wondered how the locations of these visual maps might correspond to previous anatomical designations of the lateral frontal cortex and posterior parietal cortex. To address this, we transformed and spatially aligned the maximum probability map of visual topography (PMVT; *Wang et al., 2015*) and the multi-modal parcellation of brain areas (MMP; *Glasser et al., 2016*) to each subject's native space. We were then able to compare the overlap of each subject's frontal and parietal topographic areas with these maps. *Figure 8* depicts the results of the overlap. Our IPS0-IPS3 maps tend to overlap with maps from both of these sources. However, the correspondence between specific areas is systematic neither across hemispheres nor across subjects. For instance, IPS3 from PMVT has some overlap with subject 2's left hemisphere IPS3, but not with any of the other subjects' hemispheres. Similarly, 'LIPd' and 'LIPv' from the MMP show little correspondence to our retinotopically defined IPS visual maps. In the lateral frontal cortex, we see better correspondence between our sPCS2 maps and the 'FEF' areas designated in the PMVT and MMP maps. However, this correspondence is better in the left hemisphere than in the right for some reason, and it often includes sPCS1 and misses large portions of sPCS2. MMP areas 6a and 6d typically encompass most of our sPCS1, which mostly straddles these two areas, especially in the left hemisphere.

## Discussion

Using novel procedures, we precisely characterized the topographic organization of visual field maps in human frontoparietal cortex, including four visual field maps along the IPS and two spatially tuned regions along the PCS. Each of these maps contains a representation of the full range of polar angles in the contralateral visual field; they are topographically, not simply contralaterally, organized. By combining pRFs across voxels within visual maps, we demonstrate that these maps tile the complete contralateral hemifield in an orderly manner. As expected, the pRF sizes in frontoparietal cortex are larger than those in early visual cortex. We also demonstrate previously unreported representations of eccentricity along the precentral sulcus, and replicate previous findings of eccentricity gradients along all IPS (*Swisher et al., 2007*). Furthermore, we show that a spatially tuned region of the superior precentral sulcus is organized into at least two distinct visual field maps (referred to here as sPCS1 and sPCS2), each representing the entire contralateral visual field in an orderly manner. Interestingly, the visual maps in both parietal and frontal cortex are organized into clusters of polar angle maps sharing the dimension of eccentricity, similar to visual cortex (*Wandell et al., 2005*). Two maps in frontal cortex and two pairs of maps in parietal cortex form clusters of polar angle gradients that share a foveal to peripheral representation. Together, these data clearly describe the topographic structure of the visual maps in human frontoparietal cortices.

### Frontoparietal cortex, like visual cortex, is organized into clusters

We demonstrate that visual field maps in parietal and frontal cortex are organized into map clusters, similar to the organization found in early visual cortex. Visual cortex is composed of four to five distinct visual field map clusters (*Wandell et al., 2007*). Individual maps in these clusters may perform similar, yet distinct computations that together form a larger processing unit. Their close proximity and short-range connections make information processing more efficient. We propose the existence of at least three clusters along human IPS and PCS. Following the pattern starting with V3AB, we find two pairs of angle gradients sharing a confluent foveal-to-peripheral representation: IPS0/IPS1 and IPS2/IPS3. Only one other study has reported a graded representation of eccentricity in the human parietal cortex. Consistent with our findings, *Swisher et al. (2007)* reported an eccentricity representation shared by IPS0 and IPS1, where the fundus of the IPS contained a foveally responsive area and then parafoveal-to-peripheral responses progressed medially up the sulcus, wrapping around the gyrus towards the medial wall. We report the same location of the fovea within the IPS0/1 cluster and the same mediolateral orientation. Our results not only replicate those of *Swisher et al. (2007)* but also extend them in an important way. We find the same pattern of an eccentricity representation that we found in the IPS0/1 cluster, but duplicated again more anteriorly within the IPS2/3 cluster. Therefore, both clusters have the same mediolateral organization of eccentricity. Why has this map structure gone unreported in the numerous past studies? We can only

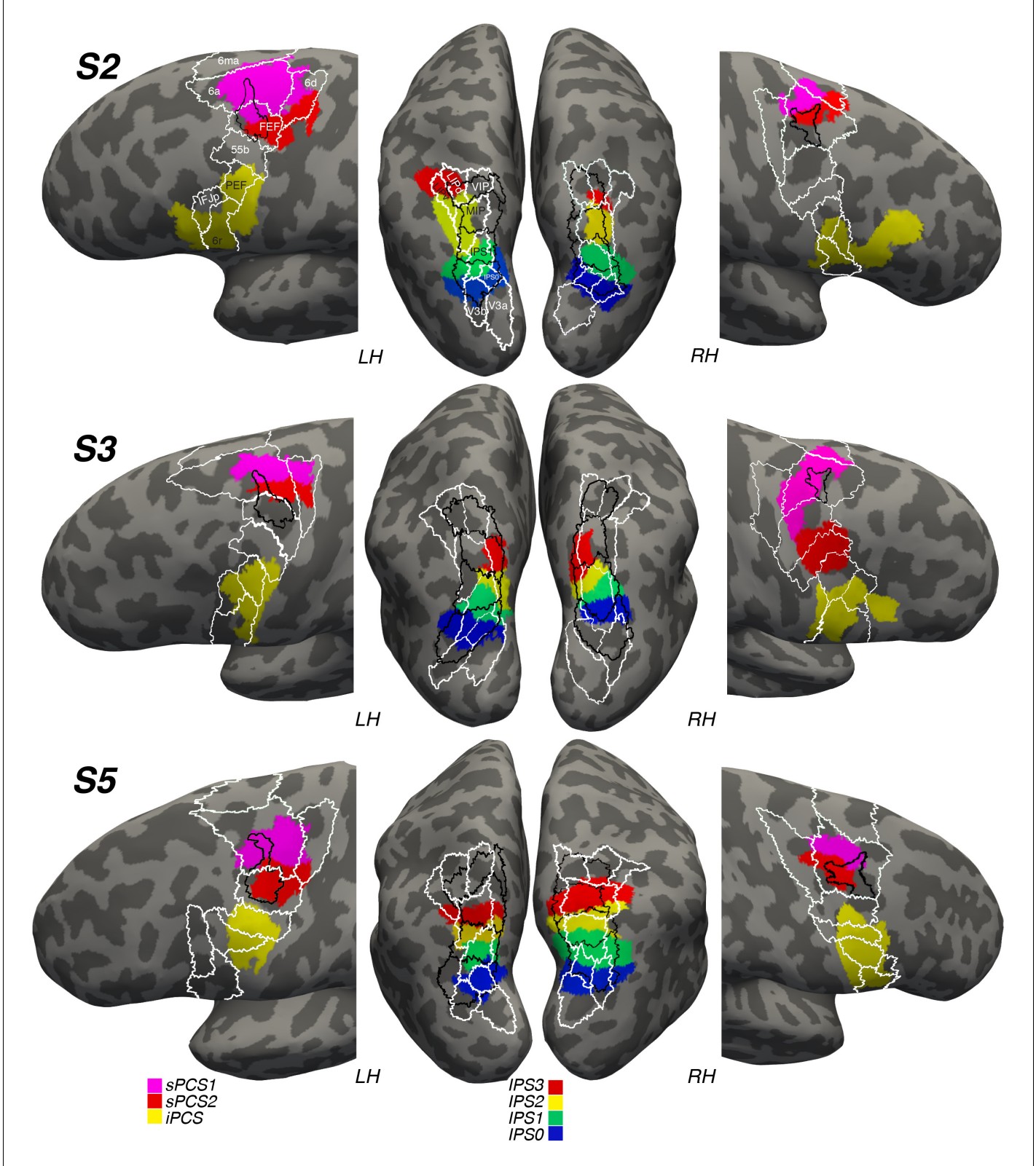

**Figure 8.** Visual maps in reference to other anatomical designations. We projected a probabilistic map of visual topography (PMVT; *Wang et al., 2015*; black outlines) and a multi-modal parcellation of brain areas (MMP; *Glasser et al., 2016*; white outlines) onto the brains of our subjects to compare these locations with those of our retinotopically defined visual maps. The PMVT includes IPS0-3 from posterior to anterior in parietal cortex, and 'FEF' in

*Figure 8 continued on next page*

*Figure 8 continued*

frontal cortex. The MMP covers the entire cortex, but here we only show the areas near or overlapping our visual maps, labeled on the images for Subject 2 (S2).

speculate that perhaps the type of task matters. For example, *Swisher et al. (2007)* used passive visual stimulation, whereas we used a demanding task that required subjects to sweep their attention across the visual field. We believe that these demands led to our main discovery. In the frontal cortex along the PCS, we find another map cluster in the superior portion (sPCS1/sPCS2) that shares a common foveal representation at the border of the two maps.

Although the non-human primate brain is typically investigated as a model of the human brain, it is important to note that the stimulus selectivity and the number and organization of visual field maps are not identical in human and non-human primates (*Tootell et al., 1997*; *Winawer et al., 2010*; *Wandell and Winawer, 2011*), and probably differ even more substantially in frontoparietal cortex. For example, although human IPS contains the putative homologue of the monkey lateral intraparietal area (LIP), human IPS houses at least four individual visual field maps (IPS0, IPS1, IPS2, IPS3) organized into two clusters (IPS0/IPS1 and IPS2/IPS3). Macaque LIP, however, appears to contain only a single visual field map, with no consistent organization of eccentricity (*Blatt et al., 1990*; *Ben Hamed et al., 2001*; *Arcaro et al., 2011*) (but see *Patel et al. (2014)*. It is likely that some of these inconsistencies between species are the result of evolution and the superior cognitive capabilities of humans (*Rilling, 2006*; *Passingham, 2009*), but it is also possible that additional macaque visual maps remain undiscovered. Although labs have mapped visual areas in non-human primates using traditional phase-based retinotopic methods (*Kolster et al., 2009*; *Patel et al., 2010*; *Savaki et al., 2010*; *Arcaro et al., 2011*), it is entirely possible that computational neuroimaging approaches, like those used here, might reveal additional visual field maps in non-human primates and might, in turn, help bridge translational research.

How and why the brain contains so many redundant maps of visual space are longstanding puzzles (*Barlow, 1986*). Certain maps, such as V1 and MT, may serve to anchor the development of other visual field maps towards clusters (*Rosa, 2002*). Perhaps the map cluster in early visual cortex acts as a developmental blueprint for replication in frontoparietal cortex (*Rosa, 2002*; *Buckner and Krienen, 2013*). Indeed, the topographic development of visual areas without retinal input mimics the organization of earlier visual areas (*Rosa and Tweedale, 2005*; *Bourne and Rosa, 2006*). Therefore, we suggest that visual field map clusters in IPS and PCS, operating like those in V1 and MT in sensory cortex, may serve as organizational anchors in the development of frontoparietal cortex.

## New visual field maps within the PCS form a visual field map cluster

We report a new subdivision of the human superior PCS consisting of two distinct angle maps (sPCS1 and sPCS2) that share a confluent foveal to peripheral representation, thereby forming a visual field map cluster. Each map contains a full representation of the contralateral visual field. A reversal in the polar angle demarcates the border between sPCS1 (LVM to UVM) and sPCS2 (UVM to LVM). The foveal representation shared by both maps sits at the intersection of the sPCS and the superior frontal sulcus. Together, these observations indicate that the eccentricity gradient and the angle gradient are not perfectly orthogonal, differing from V1, V2, and V3. However, the two gradients are not perfectly parallel either, allowing the area to represent the entire contralateral hemifield. This is similar to the observation in LO1 and LO2, which also deviate from orthogonality between the angle gradient and eccentricity gradient, but nonetheless appear to each represent a full hemifield (*Larsson and Heeger, 2006*). The organization of the sPCS into two maps, although never before reported in humans, aligns closely with recent topographic mapping results in non-human primates (*Savaki et al., 2015*) and with the results of studies of functional connectivity and domain specialization in humans (*Power et al., 2012*; *Wig et al., 2014*; *Michalka et al., 2015*).

The sPCS is the putative homologue of the monkey frontal eye fields (FEF) (*Blanke et al., 1999*) (but see *Schall et al., 2017*]), which reside in the arcuate sulcus of the macaque (*Bruce et al., 1985*). Low-level electrical stimulation of macaque FEF neurons reliably elicits saccades to particular locations in the visual field (*Robinson and Fuchs, 1969*; *Bruce et al., 1985*). Despite this well-known property of FEF neurons, the topographic organization of macaque FEF is poorly understood. For

instance, studies have revealed a coarse gradient of saccade amplitude in which small saccades are represented in the ventral portion of the FEF, and increasingly larger saccades are represented progressively more dorsally. However, the reported correspondence between stimulation site and saccade direction does not form a clearly organized map of the visual field (*Robinson and Fuchs, 1969*; *Bruce et al., 1985*). Two possibilities exist to explain the observed discrepancy between the organization of the macaque FEF and the organization of the sPCS that we describe here. First, it may be that the human homologue of the macaque FEF has expanded to contain multiple maps and a more systematic topographic organization. Such differences should be expected given the 25 million years of evolutionary divergence between monkeys and humans (*Blair Hedges and Kumar, 2003*). Second, the coarse spatial resolution of neuroimaging may be better suited than precise electrical stimulation when quantifying large-scale topographic organization. A recent study suggests that this may be the case. *Savaki et al. (2015)* imaged the distribution of metabolic activity in the macaque FEF and discovered two topographic maps of saccades to the contralateral visual field. Similar to our findings, the maps were separated by the vertical representation of space and each contained a visuomotor map of the entire contralateral visual field. The dorsal part of the arcuate sulcus contained a representation of the LVM, and progressed ventrally to a representation of the UVM. The visuomotor maps in the monkey arcuate sulcus progressed along the rostral-caudal axis, whereas the maps that we observed progressed along the dorsal–ventral axis. It is possible that this is simply a difference between the species, as other cortical (*Orban et al., 2004*) and subcortical (*Arcaro et al., 2015*) visual field map clusters are organized differently in the two species.

Our results help to shed new light on a claim that maps in frontal cortex are organized differently than maps in parietal or visual cortex (*Silver and Kastner, 2009*). The primary evidence supporting this conclusion was that the same point in space was represented multiple times in the sPCS, whereas a given point in space was represented only once in a given map in parietal or visual cortex. We argue that this apparent discrepancy stems from treating the sPCS as a single map, rather than as two maps (sPCS1 and sPCS2) as we propose here. Indeed, if one were to combine any two smaller maps in parietal or visual cortex, the same point in space would be represented more than once. Although not discussed in the original papers, a few subjects appear to have an angle reversal in sPCS that might constitute two distinct maps rather than one (figure 8 in *Kastner et al. (2007)*; figure 4 in *Hagler and Sereno, 2006*). In this study, we used eccentricity gradients derived from pRFs to identify a foveal representation in sPCS. This, in turn, was critical to revealing the structure of sPCS1 and sPCS2. We believe that the pRF-modeling technique that we used here allowed us to measure topographic organization across all subjects more reliably.

Recent functional connectivity studies have observed two distinct clusters of connectivity patterns exactly at or near the junction of the PCS and superior frontal sulcus (*Yeo et al., 2011*, *2014*). Using functional connectivity, *Wig et al. (2014)* accurately identified the border between V1 and V2. More recently, researchers have shown that both sPCS and iPCS are divided into functional clusters that have a preference for visual or auditory space (*Michalka et al., 2015*). Combining structural, functional, and resting-state MRI data from the Human Connectome Project, a semi-automated algorithm produces a reliable multi-modal parcellation (MMP) of the human cortex (*Glasser et al., 2016*). Although we did not collect the data needed by the algorithms to parcellate the cortical areas in our subjects, several areas from the parcellation averaged over many subjects overlapped the frontal and parietal visual field maps that we identified here. Similarly, the probabilistic map of visual topography (PMVT; *Wang et al. [2015]*), based on topographic mapping of 53 humans, also overlapped significantly with our visual field maps. However, neither of these mapping schemes showed good correspondence between *specific* areas across our subjects (*Figure 8*). The reason appears to be large individual differences in the functional organization of these topographic maps, across subjects and even across hemispheres within a subject. Nonetheless, future work must determine within individual subjects how sPCS1 and sPCS2, for example, might differ in terms of resting-state connectivity, diffusion-based tractography, myelination, and functional specialization. This would probably provide great insight into the unique computations performed within the visual field map cluster in frontal cortex.

It is impossible to understand how a complex system processes information without first understanding how it represents information. Sensory systems are by far the most characterized systems in all of neuroscience, primarily due to our understanding of how they represent information and our ability to use that knowledge to build models of the computations performed on those

representations. This feat has proven far more difficult in higher-order cortical areas, as the further away a neuron gets from sensory receptors, the less that neuron's firing rate appears to correlate with quantifiable elements of the external environment. By identifying how higher-order areas, namely frontoparietal cortex, represent information, our current findings provide hope for understanding how higher-order brain regions process information and contribute to cognition and behavior.

## Materials and methods

Following standard practices of reproducible research, the data (*Mackey et al., 2017*) and software (*Winawer et al., 2017*) are publicly available.

### Subjects

Five neurologically healthy individuals (1 female, mean age 33, age range 23–45) with normal or corrected-to-normal vision took part in the study. All subjects gave written informed consent before participating. All procedures were approved by the human subjects Institutional Review Board at New York University. Each subject completed one scanning session consisting of nine experimental runs.

### MRI acquisition

MRI data were collected using a 3T head-only scanner (Allegra; Siemens) at the Center for Brain Imaging at New York University. Images were acquired using a custom four-channel phased-array (NOVA Medical, Wilmington, MA ,USA) placed over lateral frontal and parietal cortices, and a four-channel phased-array placed beneath occipital cortex. Volumes were acquired using a T2*-sensitive echo planar imaging (EPI) pulse sequence (repetition time [TR], 2000 ms; echo time, 30 ms; flip angle, 75°; 31 slices; 2.5 mm x 2.5 mm x 2.5 mm voxels). T1-weighted anatomical images were collected at the beginning of each scanning session using the same slice prescriptions as for the functional data. These were used to align the functional volume to a high-resolution, whole-brain anatomical scan. High-resolution T1-weighted scans (1 mm x 1 mm x 1 mm voxels) were collected for registration, segmentation, and display.

### Topographic mapping procedures

Observers performed a difficult discrimination task that required covertly attending to stimuli within bars of different widths that swept across the visual field in different directions. The total visual field was confined to a square, 24 deg on a side, with fixation in the center of the square. The length of the bar aperture was 24 degrees, and the width subtended 1, 2, or 3 degrees of visual angle. One of the three widths was used in any given 5 min scan (similar to *Winawer et al. [2010]*). Each bar aperture was split into three equal rectangular patches along its length. For example, a bar that swept from right to left was split into a top patch, center patch, and bottom patch. A bar that swept from top to bottom was split into a left patch, center patch, and right patch.

The bar aperture swept slowly but discretely across the visual field, from one end to the other. The steps were synchronized to the MRI acquisition (one step every 2 s), and the step size was 1.6 degrees. Each bar position defined one 2-s trial. For each trial, we asked subjects to select which of the two flanking patches of moving dots matched the direction of motion in the center patch. The dot motion in the center patch was 100% coherent so that its direction was unambiguous. The motion was along the length of the bar (up or down for vertical bars sweeping horizontally, and left or right for horizontal bars sweeping vertically). The direction of motion in one of the two flanking patches was matched to the center patch, and in the other flanking patch was opposite. In order to keep the discrimination task difficult, we used a two up one down staircase on the coherence value for the moving dots in the flanker patches.

Depending on bar size, each patch contained either 124 (1 degree bar), 248 (2 degree bar), or 372 (3 degree bar) dots (each 1/10 degree in size) moving at 1.6 degrees per second. The dot positions updated 60 times per second. For the flanker patches, the set of coherent dots was randomly re-selected on each frame update, so that no single dot moved continuously in one direction throughout a trial. Dots that were not coherent disappeared and were redrawn in a random location within the aperture in the subsequent frame. Stimuli were generated in MATLAB with the MGL

toolbox and displayed on a screen in the bore of the magnet. Subjects viewed the display via a mirror mounted on the RF coil. Behavioral responses were recorded using a button box.

## MRI preprocessing

T1-weighted anatomical scans were automatically segmented using Freesurfer (*Dale et al., 1999*). All fMRI analysis was performed using the open-source Matlab tool. The first three volumes of each functional run were removed to allow magnetization to reach a steady-state. Subsequent volumes were slice-time and motion corrected using tools made available by Kendrick Kay (https://github.com/kendrickkay/preprocessfmri). Data were then aligned to each individual subject's T1-weighted anatomical image using a combination of vistasoft tools (*Winawer et al., 2017*) and Kendrick Kay's align toolbox (*Kay, 2017*). All subsequent fMRI analysis, including pRF analysis, was done using vistasoft. Functional scans for each individual experimental bar size (1, 2, and 3 degrees) were averaged together separately. Cortical surfaces were reconstructed at the gray/white matter border, and functional data (EPI time series) were projected to the gray matter voxels in the whole-brain anatomy using trilinear interpolation. Data visualization projected model parameters from the gray voxels to a smoothed 3D mesh or flattened cortical representation.

## PRF analysis

We modeled response amplitudes for each voxel using a modified version of the pRF model described by *Dumoulin and Wandell (2008)* that incorporates a static power-law nonlinearity to account for nonlinear compressive spatial summation (CSS model, *Kay et al. [2012]*). This model allows us to estimate an individual voxel's receptive field center and size. Typically, the pRF model consists of an isotropic Gaussian with four parameters: position (*x,y*), size (*σ*), and amplitude (*β*). The CSS model we employed adds an additional parameter, an exponent (*n*). This model is expressed formally as:

$$r(t) = \beta[S(x,y)G(x,y)dxdy]^n \tag{1}$$

where *r(t)* is the voxel's predicted response, *S* is the binary stimulus image, and *G* is an isotropic Gaussian expressed as:

$$G(x,y) = e^{-\frac{(x-x_0)^2+(y-y_0)^2}{2\sigma^2}} \tag{2}$$

The original pRF fitting procedure described in *Dumoulin and Wandell (2008)* involved a two-stage fitting process: an initial coarse grid-fit followed by an exhaustive search fit using non-linear search optimization algorithms. Here, we used only the first stage (the grid fit), as it is more robust to noise and our goal was to map frontoparietal cortex where signals are much noisier and pRFs are larger than in the visual cortex. We fit model predictions to temporally decimated (2x) and spatially blurred (Gaussian kernel of 5 mm width at half height) time series. Our gridded parameters included the same set of possible values as in the solution described in *Dumoulin and Wandell (2008)*, with the addition of the power law exponent (0.25, 0.5, 0.75, or 1). We interpolated solutions for voxels not included in the spatially blurred grid fit. We excluded from further analysis voxels in which the pRF model explained less than 10% of the variance of the time series, or which had pRF centers outside the limits of our visual display (12 degrees of visual angle).

## Model comparison

In order to compare nonlinear, linear, and contralateral model solutions, we cross-validated each by systematically solving the model on two-thirds of the data, and tested the model on the remaining one-third. As we used three different bar sizes in the experiment, each training data set contained six runs, with each bar size represented twice. The test data set included one run of each bar size. This was done three times so that all iterations of data shuffling were under the constraint of having each bar size equally represented in a training or test sample were satisfied. Accuracy was defined as the coefficient of determination, or variance explained by the model, and averaged for each model across the three different cross-validation iterations.

$$ve = 1 - \frac{\sum (prediction - data)^2}{\sum data^2} \tag{3}$$

## Laterality index

We calculated a laterality index for each voxel based on the $x$ and pRF size ($\sigma$) parameters of the pRF model, as performed in previous work (*Sheremata and Silver, 2015*). The only difference is that we define pRF size as $\sigma/\sqrt{n}$ rather than just $\sigma$ due to the compressive spatial summation (*Kay et al., 2013*).

$$\frac{1}{\pi} \oint_{\frac{x_0}{\sqrt{2}\sigma^{-n}}}^{\infty} e^{-g^2} dg \tag{4}$$

We also subtracted the lateralization index from 1 for left hemisphere voxels in order to compare between left and right hemispheres. This led to lateralization index values that were between 0 (completely ipsilateral) and 1 (completely contralateral).

## Test-retest analysis

Two subjects performed an additional scanning session in order to investigate the reliability of visual field map structure in frontoparietal cortex across sessions. The MRI acquisition and topographic mapping procedures for this second session were identical to those for the first session, but the behavioral task had one subtle difference. While the original session had short periods when no stimuli were presented at the beginning and end of each run, retest sessions had intermixed short blank periods during each run. These blank periods lasted 12 s, and were presented after each 48 s period of sweeping bar stimuli, similar to the traditional pRF mapping stimulus described in *Dumoulin and Wandell (2008)*.

## Probabilistic maps and multi-modal parcellation

In order to compare the locations of areas from the maximum probability map of visual topography (PMVT; (*Wang et al. [2015]*)) and the multi-modal parcellation of human cortex (MMP; (*Glasser et al. [2016]*)), we first downloaded each of the maps. Each of these are available aligned to freesurfer's fsaverage (*Dale et al., 1999*). Using freesurfer command line tools, we then simply transformed each of these surfaces to each individual subject's native anatomical space. For our purpose here, we focus on the maps along the lateral frontal cortex and the posterior parietal cortex.

# Acknowledgements

This work was supported by National Institutes of Health Grants R01 EY016407 to CEC and R00 EY022116 to JW, and by a National Science Foundation Graduate Research Fellowship Program award to WEM. We thank Lila Davachi, Martin Paré, and David Heeger for comments on early versions of the manuscript.

# Additional information

### Funding

| Funder | Grant reference number | Author |
| --- | --- | --- |
| National Institutes of Health | R01 EY016407 | Clayton E Curtis |
| National Institutes of Health | R00 EY022116 | Jonathan Winawer |
| National Science Foundation | Graduate Student Fellowship | Wayne E Mackey |

The funders had no role in study design, data collection and interpretation, or the decision to submit the work for publication.

## Author contributions
WEM, Conceptualization, Data curation, Formal analysis, Validation, Visualization, Methodology, Writing—original draft, Writing—review and editing; JW, Conceptualization, Data curation, Software, Formal analysis, Supervision, Funding acquisition, Validation, Investigation, Visualization, Methodology, Writing—original draft, Project administration, Writing—review and editing; CEC, Conceptualization, Data curation, Formal analysis, Supervision, Funding acquisition, Validation, Investigation, Visualization, Methodology, Writing—original draft, Project administration, Writing—review and editing

## Author ORCIDs
Wayne E Mackey, http://orcid.org/0000-0002-1577-9235
Jonathan Winawer, http://orcid.org/0000-0001-7475-5586
Clayton E Curtis, http://orcid.org/0000-0003-0702-1499

## Ethics
Human subjects: All subjects gave written informed consent before participating. All procedures were approved by the human subjects Institutional Review Board at New York University.

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
