## [Decision Letter]

Thank you for submitting your article "Visual field map clusters in human frontoparietal cortex" for consideration by *eLife*. Your article has been reviewed by two peer reviewers, and the evaluation has been overseen by Jack Gallant as the Reviewing Editor and David Van Essen as the Senior Editor. The following individuals involved in review of your submission have agreed to reveal their identity: Alyssa Brewer (Reviewer #1); David Somers (Reviewer #2).

The reviewers have discussed the reviews with one another and the Reviewing Editor has drafted this decision to help you prepare a revised submission.

Summary:

The reviewers believe that this is an important and worthwhile study that unnecessarily skimps on some of the statistical analyses. The suggestions of the reviewers are aimed at these issues.

Essential revisions:

1) Please rerun the pRF analysis without smoothing, as requested by Reviewer 1. The Reviewing Editor (Jack Gallant) believes that smoothing should be avoided if possible and only done when absolutely necessary, but would be satisfied if the smoothed data was retained as long as the unsmoothed data is also provided.

2) Please undertake the test-retest analysis suggested by Reviewer 2. The Reviewing Editor believes that it would be preferable to use test-retest procedures everywhere. However, Reviewer 2 is most concerned about the anterior parietal findings. It is felt that if the authors cannot do the test-retest, they should back off on their parietal claims because the result appears to be somewhat inconsistent across the small group of subjects used here.

3) Please more fully analyze the iPCS as suggested by Reviewer 2.

4) Please address the Swisher IPS results mentioned by Reviewer 2, correcting the description and improving the analysis and interpretation of the IPS maps if possible.

5) Either the authors should provide more analysis of the sPCS1/2 maps as suggested by Reviewer 2, or they should provide a more extensive discussion of these maps and potential confounds (e.g., RF size of the constituent neurons).

6) The Reviewing Editor and Senior Editor also suggest that some discussion be added to evaluate the results in the light of the recent Glasser et al. study multi-modal mapping study.

7) Although the reviewers agree that the sPCS1/sPCS2 result is solid (apart from the smoothing issue), they request that you provide further analysis of the orientation of the eccentricity and polar angle axes.

*Reviewer #1:*

This manuscript presents very detailed and informative demonstrations of visual field maps (VFMs) in frontal and parietal cortices. Eccentricity representations have been particularly difficult to measure in these regions of cortex; previous delineations have relied nearly exclusively on polar angle gradients with weak or nonexistent eccentricity gradients. The pRF modeling method presently employed (as opposed to previous phase-encoded strategies) likely played a strong role in allowing the authors to delineate the two parietal VFM clusters (IPS-0/IPS-1 and IPS-2/IPS-3) and the frontal cluster (SPCS). It is likely that previous reports of IPS-0, 1, 2, and 3 did not observe eccentricity gradients with enough accuracy to delineate whether those VFMs should be grouped into separate clusters or not, which increases the impact of the present work. The report of the frontal cluster, SPCS, with its two constituent VFMs, is the most impactful aspect of the present work, because it demonstrates that both retinotopic organization of VFMs and cluster organization between VFMs is maintained throughout the visual hierarchy rather than simply at the lower levels in visual cortex. The authors should also be commended for their focus on and presentation of extensive individual subject analyses; many previous reports depend extensively or exclusively on anatomically group-averaged datasets, which do not correctly align VFMs with one another across subjects.

However, there is one potentially problematic aspect of the present analysis, which may undercut the impact of the results. As the authors describe, they use only the "grid fit" of the pRF modeling method, which as they state, is a "coarse fit [that] was solved on time series that were spatially blurred (approximating a Gaussian kernel of 5 mm width at half height) and temporally decimated (2x)". In other words, it appears that the data were smoothed spatially and temporally, which in the original pRF modeling method is done simply to save computational resources and accelerate processing time (Dumoulin & Wandell, Neuroimage 2008). This "grid fit" was not intended to be used as the final fit. Previous reports of phase-encoded methods that employed spatial or temporal smoothing came to inaccurate conclusions about the organization of visual cortex (e.g., Hasson et al., Neuron 2002; Hadjikhani et al., Nature Neurosci 1998), which were later revised by employing better methods that did not rely on such smoothing (Larson & Heeger, J Neurosci 2006; Brewer et al., Nature Neurosci 2005). While it may be an acceptable choice to employ only the smoothed "grit fit" of the pRF method in this case, perhaps to compensate for other suboptimal or difficult-to-measure aspects of the dataset, it is not the standard pRF analysis. It is appropriate that the authors did not bury this information exclusively in the methods section of the article, but the authors should further clarify that this is not simply a choice between equally accurate pRF modeling options.

Overall, this is very well-written, clearly presented, and convincing manuscript with important implications for the fundamental organization of visual processing across cortex.

Reviewer #2:

This manuscript describes the findings of fMRI experiments that investigated the visual field representations of multiple visually responsive regions in frontal, parietal and occipital lobe of humans. Although this broad topic has been investigated multiple times over the last two decades, there remain important unanswered questions and this manuscript is poised to make a significant contribution to this literature. The key contribution of this manuscript is the presentation evidence for multiple visual field representations in superior pre-central sulcus, a putative human homolog of the Frontal Eye Fields. The manuscript also includes useful information about visual eccentricity representations in the intraparietal sulcus. Moreover, the manuscript describes methodological advances that may be of interest to many other laboratories. There are some concerns about some of the secondary claims and discussion points, but overall this manuscript has considerable merit.

The manuscript describes the results of visual mapping experiments performed on five subjects. This is a rather low N for fMRI studies, but not out of line with several prior visuotopic mapping studies. If all subjects show similar cortical organization properties, N=5, is sufficiently compelling to this reviewer. However, when there is substantial variability across subjects, N=5 leaves open many questions. Specifically, it can be unclear if this variability reflects true variations in functional organization across individuals or simply noisy data. One approach to address this is a test-retest analysis of visual field representations across scans. Such test-retest data, which is considered the 'gold standard' for claims of new visual mapping results, is not included in the manuscript.

The authors report the existence of two adjacent, but distinct visuotopic maps near the intersection of the superior frontal sulcus and the precentral sulcus. This region is commonly referred to as the superior precentral sulcus (sPCS) in visual attention studies and closely corresponds to the location of the human frontal eye fields. The analysis of five subjects reveals a clearly repeated pattern of polar angle mapping and eccentricity mapping. The regions lie in the same location across subjects, the borders between the regions are consistent across subjects, and both the polar angle map orientations and eccentric map orientations are largely consistent across subjects and hemispheres. These maps, along with the supporting analysis, are largely compelling regarding the existence of what the authors call sPCS1 and sPCS2.

Prior work from the Kastner, Silver and Sereno labs, as well as work by the senior author have revealed evidence for visual representations in sPCS; however, the prior evidence has been of coarse resolution, little better than showing preference for contralateral over ipsilateral visual targets, and has only pointed to a single visual region in sPCS. Thus, claims for sPCS1 and sPCS2 are novel and significant.

Although the sPCS1/2 story looks solid, the manuscript also addresses other brain regions and here there are issues and inadequacies that should be addressed in a revision.

Major Concerns:

1. The visually sensitive region in the inferior precentral sulcus (iPCS) is given inconsistent treatment across the manuscript. As with sPCS, several prior publications have reported greater visual sensitivity for contralateral than ipsilateral visual stimuli and at least one prior publication (Hagler & Sereno, 2006) reports consistent polar angle map representations in this region. In the present manuscript, iPCS is not given the full analysis given to sPCS, but oddly shows up in a broad cross-area comparison in the final figure (Figure 6). The amount of evidence presented for iPCS is insufficient to warrant inclusion in this manuscript. Either iPCS should be analyzed in the same detail as sPCS or it should be explicitly stated that no original claims are being made for this region (and any such claims should be excised). It would appear that the analysis of iPCS may suffer from the low N of this study.

2. The presentation of visual mapping results in intraparietal sulcus has both strengths and weaknesses. The existence of these regions and the organization of polar angle map representations in these regions is now long-established. The primary potential contribution of the present work to our understanding of IPS is the report of visual eccentricity maps and the report of two visual field map clusters, one at the IPS0/IPS1 border and the other at the IPS2/IPS3 border. The title of the manuscript, 'visual field map clusters in human frontoparietal cortex' suggests that the authors believe that this, along with the frontal lobe findings are the key points of the paper.

At this point in the review, I must 'out' myself (Somers) as the senior author on a prior visual mapping study of human intraparietal cortex (Swisher et al., 2007). In Figure 4 of that paper, we display eccentricity maps that appear largely consistent with the IPS eccentricity maps and two visual field clusters shown in Figure 3 and Supplement Figure 2 of the present manuscript. There is an important distinction to be made between the IPS0/1 cluster and the IPS2/3 cluster, so I will treat them separately.

We reported 'we reliably find a continuous gradient of eccentricity response phase along the IPS0/IPS1 border. This gradient reveals a mediolateral eccentricity representation, with a laterally position foveal representation in the fundus of the IPS, moving through parafoveal representations on the medial sulcal wall to representations of the peripheral extent of stimulation on the adjacent gyrus.' And go on to state that this organization 'defines a map complex or cluster by virtue of the confluent foveal representation.' In regards, to the IPS0/1 border, the present manuscript seems to provide only supporting evidence for these claims. While the present IPS0/1 map cluster finding should not be considered novel, the scarcity of published IPS eccentricity maps still makes this replication a worthwhile contribution to the literature. However, the discussion in the present manuscript misstates our prior finding on two points. First, it incorrectly characterizes our statement of the fovel-to-periphery gradient as medial-to-lateral, when in fact we show and state the reverse. Secondly, the manuscript states that we found this result only in a single subject; while it is true that Figure 4 displays only one subject, the finding for IPS0/1 was robust across subjects. I request that the authors revise their discussion to correctly characterize our prior findings.

With regards, to the more anterior IPS map cluster, the Swisher et al paper was more equivocal: 'Although the representation of eccentricity within IPS3/4 is less clear, a parsimonious interpretation of the data suggests that these anterior regions continue the pattern of foveal responses laterally and peripheral responses medially seen in the more posterior IPS, likely forming a second distinct foveal representation.' The present manuscript also presents equivocal results for the more anterior IPS foveal representation – in most subject hemispheres the (IPS2/3) foveal representation appears lateral as reported in Swisher et al, but in some subject hemispheres, a foveal representation appears at the medial edge of the IPS2/3 border. Moreover, in some subjects both lateral and medial foveal representations appear. It is not clear what can be concluded from the functional organization of this more anterior fovea representation. Larger N and/or test-retest analysis might help to clarify this issue. Alternately, claims regarding this anterior foveal cluster should be more strongly tempered.

3. For the reasons stated in point 2, the parietal lobe findings reported in this manuscript and thus the claims of 'frontoparietal visual field map clusters' appear to have lesser impact than the well-founded report of distinct sPCS1/sPCS2 regions. While parietal findings still belong in this manuscript (with revisions), the authors might wish to consider revising the title of the manuscript.

4. One concern regarding the sPCS1/2 maps is that the eccentricity maps appear rather parallel to the polar angle maps, unlike in posterior regions where there is a clear orthogonal relationship between these two spatial axes. Although the discussion mentions that the two axes are 'not perfectly orthogonal,' this important issue still feels a bit swept under the rug. More quantitative analysis of the relationship between the two axes and how this varies across individuals is needed. This is an important issue as prior claims of retinotopic areas (e.g. area V8) have been attacked and eventually undermined on this issue.

---

## [Author Response]

*Essential revisions:*

*1) Please rerun the pRF analysis without smoothing, as requested by Reviewer 1. The Reviewing Editor (Jack Gallant) believes that smoothing should be avoided if possible and only done when absolutely necessary, but would be satisfied if the smoothed data was retained as long as the unsmoothed data is also provided.*

We now include the unsmoothed data as requested. Importantly for future studies attempting to map these association areas using our pRF procedures, we compared the solutions from using the grid fit, which involves smoothing, and search fit, which does not directly. In comparison, we find that the grid fit solutions cross-validate slightly better than the search fit solutions, but both are robust and there is not statistical difference between the two (Figure 7—figure supplement 1). Critically, the map structures themselves appear almost indistinguishable; the use of the grid or search fit has no discernible impact on the structure of the visual field maps. This is not terribly surprising because the search fit is seeded by the first pass grid fit solution, and constrained to solutions near the seeds. For researchers interested in these details, we now include a supplemental figure of these comparisons and more importantly, we make the model solutions for both grid and search fits for all subjects available in the public data repository. Moreover, we include more details of the fitting procedures, including rationale and results in the text comparing the two fitting procedures.

*2) Please undertake the test-retest analysis suggested by Reviewer 2. The Reviewing Editor believes that it would be preferable to use test-retest procedures everywhere. However, Reviewer 2 is most concerned about the anterior parietal findings. It is felt that if the authors cannot do the test-retest, they should back off on their parietal claims because the result appears to be somewhat inconsistent across the small group of subjects used here.*

We made some minor edits to the freesurfer segmentations of a few of the hemispheres and this cleaned up the topography in PPC somewhat. Now, 9/10 hemispheres show a clear lateral foveal to medial peripheral organization that was described in Swisher et al., 2007. The remaining hemisphere also has a lateral foveal to medial peripheral organization, but appears to have some foveal representation at the medial end of the map as well. If we consider that hemisphere to be “half” consistent, then overall 9.5/10 is pretty darn consistent. We feel that this level of consistency is no longer equivocal. Moreover, in response to the reviewers we hustled to collect more data on our Allegra system in the midst of the massive construction and renovations for our new Prisma scanner. Luckily, we were able to get additional data, which allowed us to conduct test-retest analyses on two of the subjects. We strategically rescanned the two subjects whose anterior IPS eccentricity maps were the least compelling. As shown in the new figure (Figure 7), the eccentricity, as well as angle, gradient structure are indistinguishable across the two sessions, in both PCS and IPS. Therefore, we now report highly consistent eccentricity maps along IPS and we show that they are reliable across independent test-retest sessions. Remarkably, the one subject who showed what appears to be two foveal representations in IPS2/3 in session 1 also showed this in session 2, confirming both the stability of these maps and the reliability of our measurements and models. Moreover, in the first version we mainly discussed our cross-validation tests as a formal way to compare linear, non-linear, and simple laterality models. In this revision, we additionally draw attention to the fact that these cross-validation results provide additional support for the reliability of the models.

*3) Please more fully analyze the iPCS as suggested by Reviewer 2.*

We now include plots for iPCS for each individual subject (Figure 5—figure supplement 2), similar to sPCS, as well as a more in-depth description in the manuscript. Any further differences in the amount of space dedicated to sPCS over iPCS is only due to the discovery of coherent organization in sPCS, which seems to be lacking in iPCS. For example, although there seems to be a single confluent foveal representation in iPCS, the anatomical location varies between subjects. Additionally, there seems to be no obvious consistent organization in the polar angle representation across subjects.

*4) Please address the Swisher IPS results mentioned by Reviewer 2, correcting the description and improving the analysis and interpretation of the IPS maps if possible.*

First of all, we apologize to David Somers for mistaking his claims about the IPS0/1 eccentric organization. Frankly, we just mixed up our description, which flipped the direction of his reported gradient. We now correctly describe the direction of the foveal-peripheral map and we remove the comment about only showing this in a single subject, as in the manuscript they write that it was consistent in other subjects. We also expand our discussion of these findings.

*5) Either the authors should provide more analysis of the sPCS1/2 maps as suggested by Reviewer 2, or they should provide a more extensive discussion of these maps and potential confounds (e.g., RF size of the constituent neurons).*

We have included new figures (Figure 5—figure supplement 1 and Figure 6) and analysis to address these concerns, as well as added an acknowledgement of the problem with our size estimation and suggested ways in which future studies might go about addressing this problem.

*6) The Reviewing Editor and Senior Editor also suggest that some discussion be added to evaluate the results in the light of the recent Glasser et al. study multi-modal mapping study.*

We have added discussion framing our results in relation to the Glasser et al., 2016 multi-modal parcellation (MMP) brain mapping study. Specifically, we spent some time figuring out how to transform the MMP to each subjects’ native space so that we could make a direct comparison. We did the same thing for the probabilistic maps of visual topography (PMVT; Wang et al., 2012). We have a new figure (Figure 8) that displays the overlap in our subjects. Now, we discuss the alignment between our retinotopic visual maps and those in the MMP and PMVT. In a nutshell, there is some interesting areas of overlap between all three mapping schemes, but these do not seem to be particularly consistent across participants. We acknowledge that the MMP is not an atlas, and was not designed to work on an individual based solely on anatomical landmarks. Nonetheless, readers may find it useful to “ball park” these existing maps with our frontal and parietal visual maps.

*7) Although the reviewers agree that the sPCS1/sPCS2 result is solid (apart from the smoothing issue), they request that you provide further analysis of the orientation of the eccentricity and polar angle axes.*

This was very good advice. We now further describe how the eccentricity-angle angle is not perfectly orthogonal. We also include a new figure that shows maps of visual field coverage plots in sPCS1/2 for individual subjects (Figure 6). These clearly show that despite non-orthogonality the voxels in the map tile visual space. If the angle and eccentricity maps were aligned, then one would predict that the visual field coverage plot would form a spiral of the pRF centers. As can be seen, none of the subjects show anything remotely resembling a spiral. The pRFs are regularly dispersed in mostly the contralateral VF.

*Reviewer #1:*

*[…] However, there is one potentially problematic aspect of the present analysis, which may undercut the impact of the results. As the authors describe, they use only the "grid fit" of the pRF modeling method, which as they state, is a "coarse fit [that] was solved on time series that were spatially blurred (approximating a Gaussian kernel of 5 mm width at half height) and temporally decimated (2x)". In other words, it appears that the data were smoothed spatially and temporally, which in the original pRF modeling method is done simply to save computational resources and accelerate processing time (Dumoulin & Wandell, Neuroimage 2008). This "grid fit" was not intended to be used as the final fit. Previous reports of phase-encoded methods that employed spatial or temporal smoothing came to inaccurate conclusions about the organization of visual cortex (e.g., Hasson et al., Neuron 2002; Hadjikhani et al., Nature Neurosci 1998), which were later revised by employing better methods that did not rely on such smoothing (Larson & Heeger, J Neurosci 2006; Brewer et al., Nature Neurosci 2005). While it may be an acceptable choice to employ only the smoothed "grit fit" of the pRF method in this case, perhaps to compensate for other suboptimal or difficult-to-measure aspects of the dataset, it is not the standard pRF analysis. It is appropriate that the authors did not bury this information exclusively in the methods section of the article, but the authors should further clarify that this is not simply a choice between equally accurate pRF modeling options.*

Thank you, this is a great point, as we did indeed have reasoning for choosing the grid fit but unfortunately did not sufficiently explain our reasoning in the manuscript. We have added a new cross-validation analysis comparing the search and grid fit (Figure 7—figure supplement 5B). For this analysis, both the grid fit and the search fit are cross-validated against the original, unsmoothed left-out test data. Interestingly, the grid fit is at least as accurate as the search fit for all visual areas (though the differences are not statistically significant). Although the grid fit is trained on less data, it does not suffer from reduced accuracy. We believe this is due to the search fit being more susceptible to noise, while the grid fit imposes a “prior” or inherent structure in the maps by sampling every other voxel. See above in “Essential Revisions” for a description of our response to this issue. Overall, we are pleased that the map structure is similar to the two types of solutions, and we believe that the inclusion of the search fit adds further credibility to the results and conclusions.

*Reviewer #2:*

*[…] Major Concerns:*

*1. The visually sensitive region in the inferior precentral sulcus (iPCS) is given inconsistent treatment across the manuscript. As with sPCS, several prior publications have reported greater visual sensitivity for contralateral than ipsilateral visual stimuli and at least one prior publication (Hagler & Sereno, 2006) reports consistent polar angle map representations in this region. In the present manuscript, iPCS is not given the full analysis given to sPCS, but oddly shows up in a broad cross-area comparison in the final figure (Figure 6). The amount of evidence presented for iPCS is insufficient to warrant inclusion in this manuscript. Either iPCS should be analyzed in the same detail as sPCS or it should be explicitly stated that no original claims are being made for this region (and any such claims should be excised). It would appear that the analysis of iPCS may suffer from the low N of this study.*

We now include plots for iPCS for each individual subject, similar to sPCS, as well as a more in-depth description in the manuscript. As the cross validation of our results was high in this area for all subjects, we don’t believe our failure to identify a consistent organization is due to low power. See above in “Essential Revisions” for a description of our response to this issue.

*2. […] We reported 'we reliably find a continuous gradient of eccentricity response phase along the IPS0/IPS1 border. This gradient reveals a mediolateral eccentricity representation, with a laterally position foveal representation in the fundus of the IPS, moving through parafoveal representations on the medial sulcal wall to representations of the peripheral extent of stimulation on the adjacent gyrus.' And go on to state that this organization 'defines a map complex or cluster by virtue of the confluent foveal representation.' In regards, to the IPS0/1 border, the present manuscript seems to provide only supporting evidence for these claims. While the present IPS0/1 map cluster finding should not be considered novel, the scarcity of published IPS eccentricity maps still makes this replication a worthwhile contribution to the literature. However, the discussion in the present manuscript misstates our prior finding on two points. First, it incorrectly characterizes our statement of the fovel-to-periphery gradient as medial-to-lateral, when in fact we show and state the reverse. Secondly, the manuscript states that we found this result only in a single subject; while it is true that Figure 4 displays only one subject, the finding for IPS0/1 was robust across subjects. I request that the authors revise their discussion to correctly characterize our prior findings.*

Thank you for identifying our mistake. Our updated maps, test-retest results, and rewritten description now perfectly align with Swisher et al., 2007. See above in “Essential Revisions” for a description of our response to this issue.

*With regards, to the more anterior IPS map cluster, the Swisher et al paper was more equivocal: 'Although the representation of eccentricity within IPS3/4 is less clear, a parsimonious interpretation of the data suggests that these anterior regions continue the pattern of foveal responses laterally and peripheral responses medially seen in the more posterior IPS, likely forming a second distinct foveal representation.' The present manuscript also presents equivocal results for the more anterior IPS foveal representation – in most subject hemispheres the (IPS2/3) foveal representation appears lateral as reported in Swisher et al, but in some subject hemispheres, a foveal representation appears at the medial edge of the IPS2/3 border. Moreover, in some subjects both lateral and medial foveal representations appear. It is not clear what can be concluded from the functional organization of this more anterior fovea representation. Larger N and/or test-retest analysis might help to clarify this issue. Alternately, claims regarding this anterior foveal cluster should be more strongly tempered.*

See above in “Essential Revisions” for a description of our response to this issue.

*3. For the reasons stated in point 2, the parietal lobe findings reported in this manuscript and thus the claims of 'frontoparietal visual field map clusters' appear to have lesser impact than the well-founded report of distinct sPCS1/sPCS2 regions. While parietal findings still belong in this manuscript (with revisions), the authors might wish to consider revising the title of the manuscript.*

We have made revisions to the discussion of the parietal findings to address any perceived issues with inconsistency in organization and feel the parietal results, although not as novel, are still an important part of the paper. Therefore, we wish to keep the title.

*4. One concern regarding the sPCS1/2 maps is that the eccentricity maps appear rather parallel to the polar angle maps, unlike in posterior regions where there is a clear orthogonal relationship between these two spatial axes. Although the discussion mentions that the two axes are 'not perfectly orthogonal,' this important issue still feels a bit swept under the rug. More quantitative analysis of the relationship between the two axes and how this varies across individuals is needed. This is an important issue as prior claims of retinotopic areas (e.g. area V8) have been attacked and eventually undermined on this issue.*

As now seen in the new figures, none of the subjects have axes that are orthogonal, but none are parallel either. See above in “Essential Revisions” for a description of our response to this issue.